# Environmental Drivers and Potential Distribution of *Schistosoma mansoni* Endemic Areas in Ethiopia

**DOI:** 10.3390/microorganisms9102144

**Published:** 2021-10-14

**Authors:** Keerati Ponpetch, Berhanu Erko, Teshome Bekana, Tadesse Kebede, Di Tian, Yang Yang, Song Liang

**Affiliations:** 1Department of Environmental and Global Health, College of Public Health and Health Professions, University of Florida, Gainesville, FL 32611, USA; 2Emerging Pathogens Institute, University of Florida, Gainesville, FL 32611, USA; yangyang@ufl.edu; 3Praboromarajchanok Institute, Faculty of Public Health and Allied Health Sciences, Sirindhorn College of Public Health Trang, Trang 92110, Thailand; 4Aklilu Lemma Institute of Pathobiology, Addis Ababa University, Addis Ababa 3614, Ethiopia; berhanue@yahoo.com (B.E.); teshomebekana@gmail.com (T.B.); tadesse.kebed@gmail.com (T.K.); 5Department of Microbiology, Immunology and Parasitology, School of Medicine, Addis Ababa University, Addis Ababa 9086, Ethiopia; 6Department of Crop, Soil, and Environmental Science, Auburn University, Auburn, AL 36849, USA; tiandi@auburn.edu; 7Department of Biostatistics, College of Public Health and Health Professions, University of Florida, Gainesville, FL 32611, USA

**Keywords:** environmental driver, *Schistosoma mansoni*, Ethiopia, ecological niche modeling

## Abstract

In Ethiopia, human schistosomiasis is caused by two species of schistosome, *Schistosoma mansoni and S. haematobium,* with the former being dominant in the country, causing infections of more than 5 million people and more than 37 million at risk of infection. What is more, new transmission foci for *S. mansoni* have been reported over the past years in the country, raising concerns over the potential impacts of environmental changes (e.g., climate change) on the disease spread. Knowledge on the distribution of schistosomiasis endemic areas and associated drivers is much needed for surveillance and control programs in the country. Here we report a study that aims to examine environmental determinants underlying the distribution and suitability of *S. mansoni* endemic areas at the national scale of Ethiopia. The study identified that, among five physical environmental factors examined, soil property, elevation, and climatic factors (e.g., precipitation and temperature) are key factors associated with the distribution of *S. mansoni* endemic areas. The model predicted that the suitable areas for schistosomiasis transmission are largely distributed in northern, central, and western parts of the country, suggesting a potentially wide distribution of *S. mansoni* endemic areas. The findings of this study are potentially instrumental to inform public health surveillance, intervention, and future research on schistosomiasis in Ethiopia. The modeling approaches employed in this study may be extended to other schistosomiasis endemic regions and to other vector-borne diseases.

## 1. Introduction

Schistosomiasis is a neglected tropical disease affecting more than 258 million people in 78 countries worldwide, with over 90% of human cases occurring in sub-Saharan Africa (SSA) [1]. In Ethiopia, human schistosomiasis is caused by *Schistosoma mansoni* and *S. haematobium*, with the former being dominant. Over 5 million people are estimated to be infected by *S. mansoni*, and 37 million at risk of infection [2]. Information on the distribution of endemic areas was largely based on many school- and community-based epidemiological surveys in past decades [3]. Further, new transmission foci have been constantly reported in the past years, raising concerns over potential impacts of environmental changes (e.g., climate change and water resource projects) on the disease spread [4]. There is a pressing need for improved knowledge on the distribution of endemic areas, particularly at the national scale, and underlying risk factors to inform control and surveillance programs.

Schistosomiasis transmission involves complex interactions between humans, parasites, and snail intermediate hosts in the freshwater environment [5]. Many environmental factors, such as temperature, precipitation, elevation, soil, and vegetation have been explored in relation to the distribution of schistosomiasis endemic areas, and the importance of these factors may vary with the parasite species and eco-epidemiological setting [6]. For example, rainfall and water resource development projects can facilitate the development of habitats suitable for snail intermediate hosts [7]. Temperature can affect parasite development in snail intermediate hosts and cercariae infectivity [8,9]. Elevation is an important indicator for spatial monitoring of snail-borne diseases because some snail species favor certain ranges of elevation. For example, *Biomphalaria pfeifferi,* an intermediate host of *S. mansoni*, is mostly found at an altitude of 1000 to 2000 m above sea level [10]. Soil components also play an important role in schistosomiasis transmission, as they affect land use and agriculture practices thereby affect the ecology of snail intermediate hosts [11].

Geographic information systems (GISs) and remote sensing techniques have offered powerful tools for generating, storing, processing, and visualizing data (e.g., satellite imagery-derived environmental data). Various statistical and mathematical methods have been developed to analyze these georeferenced data in order to understand the relationships between environmental factors and living species (e.g., plant, animals, or disease vectors) and predict the potential distribution of species across the landscape [12]. Ecological niche modeling (ENM) involves a class of methods that use occurrence data of species and environmental data to estimate potential species habitats [13]. In public health, ENM has been widely used to understand disease ecologies, in particular related to vector-borne diseases, by modeling the occurrence of the disease or species involved in disease transmission (e.g., hosts, vectors, and pathogens) [14]. This study aims to assess environmental determinants and predict the national distribution of *S. mansoni* endemic areas in Ethiopia.

## 2. Materials and Methods

### 2.1. Schistosomiasis Occurrence Data

A GIS database of historical *S. mansoni* infection surveys was developed from a systematic review and described in a separate paper [15]. In brief, a systematic review was conducted to assess *S. mansoni* infections (e.g., prevalence of infection, infection intensity, and distribution) in Ethiopia. The essential data were extracted from published articles and reports (e.g., gray literature), including year, study site (e.g., village/town, district/region, county), geographic coordinates of the study site, target population, sample size, number of positive individuals, prevalence and intensity of infections, and diagnostic techniques used. A total of 95 endemic localities from 62 studies were identified in the systematic review and were included in the current analysis. Additionally, 9 studies reported zero prevalence of *S. mansoni* infection, and the corresponding 9 sites were used for model validation but not for model development. The 95 endemic sites were treated as ‘occurrence’ points and were georeferenced (Figure 1) for subsequent analyses. However, some ENM techniques used in this study require a sufficient number of both presence and absence data to develop the models [16]. To follow widely accepted practices in the field of ecology, we augmented the presence data with pseudoabsence data points for the niche modeling [17,18]. Specifically, 1000 pseudoabsence data points 20 km away from the ‘occurrence’ locations were randomly generated across the country. The number of pseudoabsence data points is based on empirical evidence from a previous study [19], which suggests that the model accuracy is usually satisfactory when the ratio of presence vs. pseudoabsence reaches approximately 1:10. Choosing an area of a 20 km radius to generate pseudoabsence points is based on the resolution of environmental data (1 km^2^), allowing adequate environmental differentiation between presence and pseudoabsence locations, a criterion that has also been empirically confirmed by the field epidemiologist and parasitologist in Ethiopia (BE). The procedure of selecting pseudoabsence points was implemented in ArcGIS version 10.3.1.

### 2.2. Environmental and Ecological Data

Environmental and ecological variables used in this study are summarized in Table 1. These data were obtained from different sources. Elevation data at a resolution of approximately 250 m was obtained from CGIAR Consortium for Spatial Information (CGIAR-CSI) [20]. The Normalized Difference Vegetation Index (NDVI) data at the global scale were generated by Copernicus Global Land Service using long-term statistics over 1999–2017 [21]. The wealth index, provided by the Demographic and Health Surveys (DHS) based on data of household assets, nutrition services, healthcare, education, and other economic indicators [22], has been widely used in characterizing health and economic status [23,24,25]. Wealth index score and cluster coordinates were obtained from Ethiopia’s survey data in 2016 [26]. Data were then georeferenced and interpolated using the Empirical Bayesian Kriging technique [27]. Soil content data were obtained from the International Soil Reference and Information Centre (ISRIC) and reported as a percentage of sand, silt, and clay relative to total soil (g/100 g) [28].

Climate variables used in the analysis were obtained from two sources: WorldClim and the Statistical DownScaling Model (SDSM). The WorldClim offers a global-scale climate dataset and has been used widely in various fields such as geography, ecology, and public health [31,32,33,34]. However, the coarse resolution of global climate data from the WorldClim dataset is likely to limit its usability for studies at the local scale [35]. Moreover, the WorldClim climate data may be associated with potential biases and uncertainties when illustrating present and future climate scenarios at the local scale [36]. The second source of climate data is the SDSM, which is a statistically downscaled climate dataset at the local scale for Ethiopia, Kenya, and Tanzania, based on daily observed weather data (e.g., precipitation, minimum and maximum temperature) from 211 stations in the three countries [30]. The two datasets were developed based on different data sources and algorithms and have different resolutions and accuracies that may result in variations in associated ‘observations’ (e.g., temperature and precipitation) and in turn may impact the empirical relationships to be studied. In the preliminary investigation, we found notable differences in climate variable values between the two datasets in southwestern Ethiopia. To account for such inconsistency and uncertainty, we developed ENMs based on each dataset separately and compared the performance of corresponding models. We obtained annual average temperature and cumulative precipitation from the WorldClim (WC) dataset, which covers the period of 1970–2000, and obtained minimum and maximum temperatures and mean precipitation aggregated data between 1961 and 2005, from the Statistical DownScaling Model (SDSM) dataset.

### 2.3. Ecological Niche Modeling

An ecological niche of a species is a combination of environmental conditions that allow the species to maintain or grow its population in the long term without the need for immigration [37]. Ecological niche modeling (ENM) has been increasingly used in public health, in particular for understanding the ecology of vector-borne diseases. For example, ENM has been used to identify geographic spaces and ecological conditions that can support disease agents or promote transmission [38]. Several ENM techniques have been developed or extended to estimate the spatial distributions of species, such as the genetic algorithm for rule-set prediction (GARP), maximum entropy (MaxEnt), random forest (RF), generalized linear model (GLM), and generalized additive model (GAM) [39,40,41,42]. However, in practice, choosing a technique most suitable for a specific application remains challenging because these techniques have specific data requirements (e.g., presence–absence, presence–background, and presence-only data), and their performances heavily depend on various factors such as predictor variables, model tuning parameters, and selection of pseudoabsence or background data. [43,44]. A plausible solution to this issue is to combine multiple niche modeling techniques by using an ensemble architecture [45].

In this study, we used eight ENM techniques to analyze the data. The outputs from the eight techniques were then used to develop ensemble models for predicting the potential distribution of *S. mansoni* endemic areas in Ethiopia. An ensemble model is a machine learning approach that combines all ENM models to improve predictive performance [46,47]. Among the eight techniques, two require presence-only data (Surface Range Envelope (SRE) and MaxEnt). The remaining techniques use both presence and pseudoabsence data, including three regression-based techniques (generalized linear model (GLM), generalized additive model (GAM), and multivariate adaptive regression splines (MARS)), and three decision-tree-based techniques (Classification Tree Analysis (CTA), random forest (RF), and generalized boosting model (GBM)). Predicted occurrence probabilities from the ensemble model were projected to the distribution map [31].

### 2.4. Model Evaluation

The model evaluation consisted of two independent processes using presence and pseudoabsence data and ground truthing. For the former, all occurrence (95 sites) and pseudoabsence (1000 sites) data were randomly split into two parts with an 80:20 ratio, with 80% (training set) for training the model and the remaining 20% (testing set) for validating the predictive performance of the model. The receiver operating characteristic (AUC) was applied for model evaluation. A model with an AUC lower than 0.7 was considered poorly predictive, between 0.7 and 0.9 as moderately predictive, and greater than 0.9 as highly predictive [48]. ENM models with an AUC greater than 0.7 were then included in the ensemble model. To translate the occurrence probabilities produced by the ensemble model to a binary outcome (i.e., suitable or nonsuitable), we explored an array of threshold values (i.e., from 0.3 to 0.8) and used the cutoff point that maximized the sensitivity and specificity of the model for classifying the ecological suitability, which has been commonly used for predicting species distributions [49].

Relative importance of predictor variables was evaluated using the following procedure. A standard prediction was performed when the models were trained. Then, one of the predictor variables was dropped, and the model was refitted to make a new prediction. A correlation score was calculated between the new prediction and the standard prediction for this particular variable. This procedure of dropping and refitting was repeated for all variables to be evaluated. The resulting correlation scores were then used to evaluate the relative importance of variables in the models. If the two predictions showed a strong positive correlation, the predictor was considered of low importance for the model [50].

The second process of model evaluation involved the use of a mixed approach: a ground-truthing validation and validation using the nine sites reporting the absence of *S. mansoni* derived from the systematic review. To perform the field ground validation, 25 sites were initially considered, and among them, 16 sites in two regions, Oromia and Southern Nations, Nationalities, and Peoples’ Region (SNNP), were randomly selected for field validation during December 7–22, 2020 (conducted by K.P., B.E., and T.B.). At each selected site, a snail survey was conducted on *Biomphalaria pfeifferi* and *B. sudanica*, snail intermediate hosts of *S. mansoni.* The geographic coordinates of survey sites were also taken during the survey. Finally, these 25 localities were georeferenced and plotted on the predicted risk map for evaluating the accuracy (proportion of number of correct predictions and number of all assessments), sensitivity (proportion of number of true positive predictions correctly classified), specificity (proportion of number of true negatives correctly classified), positive predictive value (proportion of true positives and number of all positive predictions), negative predictive value (proportion of true negative numbers of all negative predictions), and balanced accuracy (arithmetic mean of the sensitivity and specificity to account for imbalance classification classes) of model predictions.

## 3. Results

### 3.1. Model Performance

Overall, all eight models exhibited good performance, with the AUC ranging from 0.72 to 0.93 for the WC dataset and 0.74 to 0.88 for the SDSM dataset (Table 2). GBM and RF had the highest AUC for both climate models. Only SRE and CTA had an AUC lower than 0.8 for both datasets. GLM, GBM, GAM, RF, and MaxEnt models using the WC dataset showed higher AUC scores than those using the SDSM dataset, whereas SRE, CTA, and MARS using the SDSM dataset had higher AUC scores than those using the WC. All models had AUC values greater than 0.7, so all models were included in the ensemble model for each climate dataset. The ensemble models using two different weather datasets showed similar AUC values, 0.956 vs. 0.960, which are indeed higher than individual models.

For the field ground-truthing validation, occurrence probabilities were predicted by all individual models as well as the ensemble models, at the 25 locations from validation data and were transformed to endemicity maps (e.g., either ‘endemic’ or ‘nonendemic’), with a threshold of 65 percent probability, and then were compared to the survey results to calculate accuracy, sensitivity, specificity, positive predictive value (PPV), negative predictive value (NPV), and balanced accuracy. The models fitted to the WC data showed an accuracy between 37.50% and 83.33%, sensitivity varied from 25.00% to 100.00%, specificity between 12.50% and 62.50%, PPV between 66.67% and 84.21%, NPV between 33.33% and 100.00%, and balance accuracy was from 40.63% to 78.13%. Ensemble and GBM shared the highest accuracy and balance accuracy at 83.33% and 78.13%, respectively. On the other hand, SRE and MaxEnt shared the lowest balance accuracy at 43.75%.

For the models fitted to the SDSM data, the accuracy varied between 29.17% and 83.33%, sensitivity between 12.50% and 100.00%, specificity between 25.00% and 87.50%, PPV between 66.67% and 84.21%, NPV between 33.33% and 100.00%, and balance accuracy between 37.50% and 84.38%. RF had the highest percentage of accuracy and balance accuracy at 83.33% and 84.38%, respectively. Like the models fitted to the WC data, MaxEnt produced the lowest accuracy and balance accuracy at 29.17% and 37.50%, respectively. In addition, the ensemble model fitted to SDSM had low performance based on the ground truth, showing accuracy, sensitivity, and balance accuracy being less than 50.00% (Table 3).

### 3.2. Relative Importance of Variables

For both WC-based and SDSM-based models, the elevation and soil properties (silt, clay, and sand) appeared to be the most important predictors. Temperature also had a considerable influence on the predictive performance of the models, regardless of which climate dataset was used. The wealth index and NDVI were of the least importance for both WC-based and SDSM-based models (Figure 2).

### 3.3. Distribution of S. mansoni Endemic Areas in Ethiopia

Figure 3 and Figure 4 showed the spatial distributions of *S. mansoni* endemic areas in Ethiopia predicted by the ensemble models as well as individual models built on the two climate datasets, respectively. General spatial patterns were similar between the two ensemble models, with slight differences existing in west Ethiopia. The SDSM-based model estimated a low probability of disease risk in the west (where previous epidemiological surveys lacked), where the WC-based model suggested extended endemic areas beyond current reported ones in this region. Overall, endemic areas (shown in red) were largely distributed in northern, central, and western Ethiopia. However, there were notable variations in the predictions by the individual models, regardless of which climate dataset was used. For example, GLM, GAM, and MARS suggested wider distributions of the disease-endemic areas than that predicted by RF. In addition, the presence–absence maps based on ensemble models showed mild differences in projections between the WC-based and the SDSM-based models not only in the west but also in the east of the country in the eastern part of the country (Figure 5).

## 4. Discussion

In this study, we conducted ecological niche modeling using *S. mansoni* occurrence data in conjunction with social and environmental data (i.e., elevation, NDVI, soil properties, and wealth index) and climatic data (i.e., temperature and precipitation) to assess the roles of these factors and predict the potential distribution of endemic areas in Ethiopia. Eight ENM techniques were used, and a resulting ensemble model was developed. This study is among the first to use the ENM approach to study environmental drivers and distribution of *S. mansoni* endemic areas in Africa. Previous studies in China [51] and South Africa [52] used the ENM approach to estimate the potential habitat of snail intermediate hosts and predicted the risk area of schistosome transmission. Our findings are potentially instrumental for applications in the public health sector, particularly for schistosomiasis surveillance and control programs.

Among the eight ENM and ensemble models developed, we found that the ensemble models had the greatest AUC scores using climate data from both sources. This was because the ensemble platform integrated the ENM models with an AUC score greater than 0.7. Higher performance of the ensemble model compared to individual ENM models, as shown in our study, was also reported in other ecological studies. For example, a study in the United States showed a potentially suitable habitat for three different fishes [53], and Dal Maso and Montecchio [54] studied the geographical distribution of ash dieback disease among trees in Europe. Results from our ensemble models showed that the majority of suitable areas for *S. mansoni* in Ethiopia were located in the West Montane and Rift Valley ecozones [15]. The findings largely agreed with reports of the high rate of infection in villages around Lake Tana and in rift valley areas near Lake Ziway and Lake Abaya reported three decades ago [55].

In alignment with other studies, our study found some key environmental and ecological factors underlying the distribution of schistosomiasis endemic areas [56]. Particularly, our results from both WC-based and SDSM-based models showed that soil properties are an important factor. This finding agrees with a previous study that reported the importance of soil components in terms of favoring the population growth of snail intermediate hosts [57]. It is interesting to note that the model performance was also influenced by elevation, which is the most influential variable in the WC-based model and the second most in the SDSM-based model. This result confirmed that elevation was an important indicator for spatial monitoring of snail-borne diseases because certain ranges of elevation favor some snail species [15]. Consequently, elevation can be used to help identify the populations most vulnerable to schistosomiasis [55]. Climate variables ranked after soil properties and elevation in terms of variable importance are also pertinent to the schistosomiasis transmission cycle. For example, temperature and precipitation were identified as important factors promoting or inhibiting the growth of parasite and snail intermediate hosts [5]. Shifts in precipitation and temperature patterns can also affect the transmission of schistosomiasis. For example, increasing water temperature may influence cercaria production and shedding from infected snails. Thus, global climate change may further influence the distribution of the disease in the future. In addition, some environmental changes from human activities such as water resource projects may create suitable habitats for intermediate hosts, allowing the disease to spread into nonendemic areas [9].

In this study, we used climate data from two sources (WC and SDSM) to account for potential uncertainties and variabilities due to availability, resolution, and accuracy of the data. As in traditional ENM analyses, we split original data into model-training and model-testing subsets to assess the accuracy of model predictions. The ensemble models based on the two data sources yielded similar predictive performance (AUC = 0.96). In addition to the traditional cross-validation, a field truthing evaluation at randomly sampled locations was also performed, which indicated high accuracy, sensitivity, specificity, PPV, NPV, and balance accuracy of the WC-based models compared to the SDSM-based models. A caveat of the field truthing is its small sample size. Therefore, the field truthing was not meant to replace but to supplement the classic approach of cross-validation.

One limitation of the study is that part of the absence data used in the study was based on pseudoabsence data. Due to the lack of disease-absence data, we generated pseudoabsence data by randomly selecting 1000 locations across the country with the condition that random points are 20 km away from occurrence locations. Although some presence-only algorithms are available now, ENM model performance comparisons show that presence–absence techniques tend to perform better [19]. However, the availability of absence data was usually limited. In this study, we adopted the commonly used approach to overcome this problem by simulating pseudoabsence data with specific criteria. Additionally, selecting a single best-fit model has certain limitations, e.g., differences in data requirement and accuracy, uncertainty of model performance, we used a multimodel ensemble platform based on eight different ENM techniques (e.g., both presence-only and presence–absence data) to predict potential endemic areas. An ensemble model is more robust than a single model and has gained increasing popularity [50]. Thus, using pseudoabsence data with a specific condition can improve the overall model performance and provide more insights compared to using present-only data.

In this study, we present a case applying ENM and ensemble modeling techniques to *S. mansoni* in Ethiopia. The techniques can be extended to other diseases, particularly related to environmentally mediated infectious diseases. Increasing availability of environmental data makes ENM techniques particularly attractive. At the global or continental scales, open-source data, such as remote sensing imagery and land use/cover, are often georeferenced and can be valuable sources of information for ENM applications [58]. It should be noted that developing a model at the local scale often needs environmental data of fine resolution, which may not be readily available and require researchers to generate necessary data and/or need to take the interactions of biotic variables into account [37]. Nevertheless, the techniques and modeling approaches have great potential for helping to understand disease occurrence and distribution and informing appropriate surveillance and control programs.

## Figures and Tables

**Figure 1 microorganisms-09-02144-f001:**
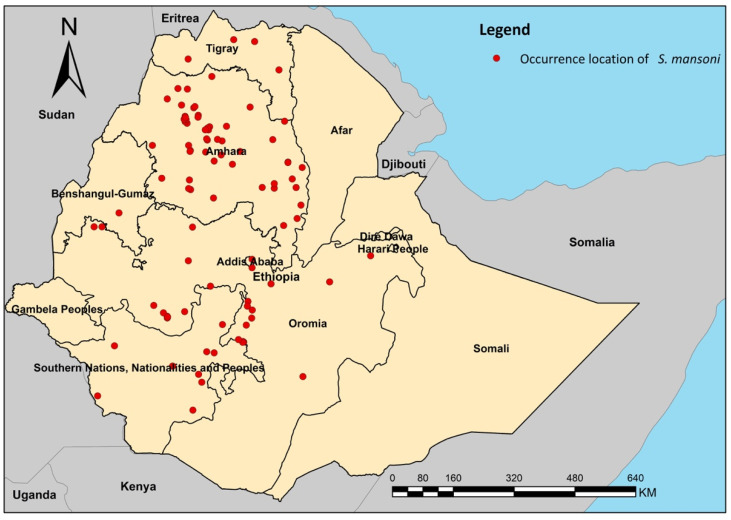
Occurrence locations of *S. mansoni* endemic areas derived from 62 surveys in Ethiopia.

**Figure 2 microorganisms-09-02144-f002:**
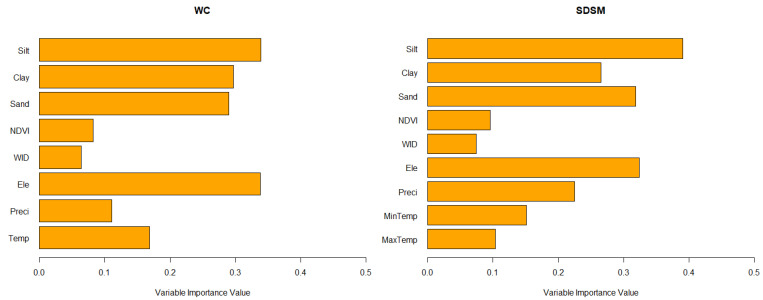
Mean variable importance from all WC-based and SDSM-based models (WID: Wealth Index, Ele: Elevation, Preci: Precipitation, Temp: Temperature, MaxTemp: Maximum Temperature, and MinTemp: Minimum Temperature).

**Figure 3 microorganisms-09-02144-f003:**
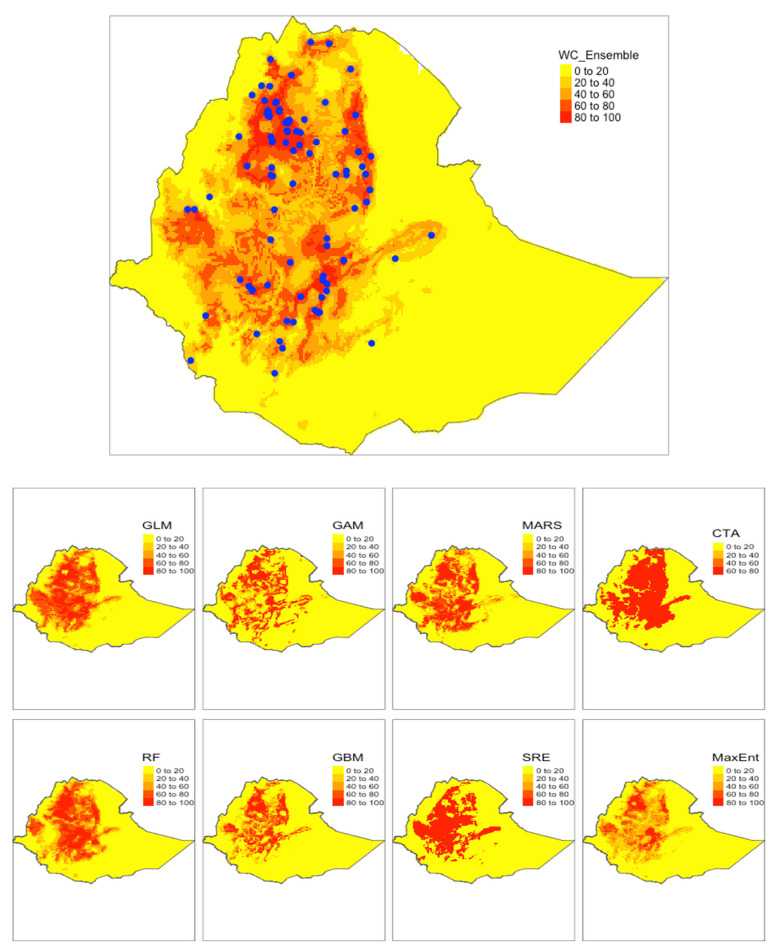
WC-based results showing potential distribution of *S. mansoni* endemic areas in Ethiopia from different models. Blue dots represent the occurrence locations. Legend shows a probability of occurrence given the environmental suitability ranging from 0 to 100 percent.

**Figure 4 microorganisms-09-02144-f004:**
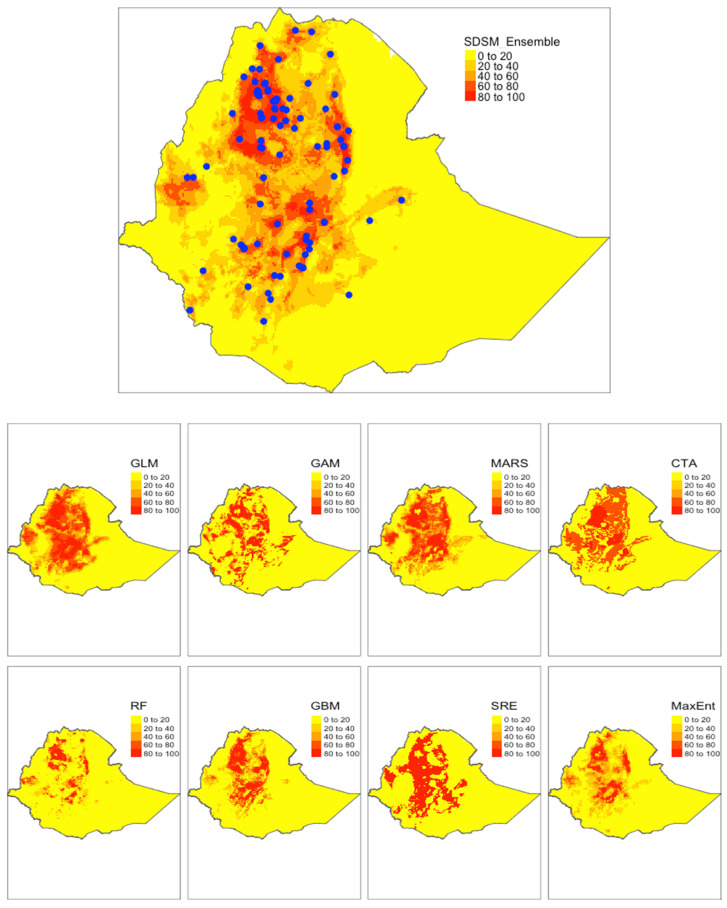
SDSM-based results showing potential distribution of *S. mansoni* endemic areas in Ethiopia from different models. Blue dots represent the occurrence locations. Legend shows a probability of occurrence given the environmental suitability ranging from 0 to 100 percent.

**Figure 5 microorganisms-09-02144-f005:**
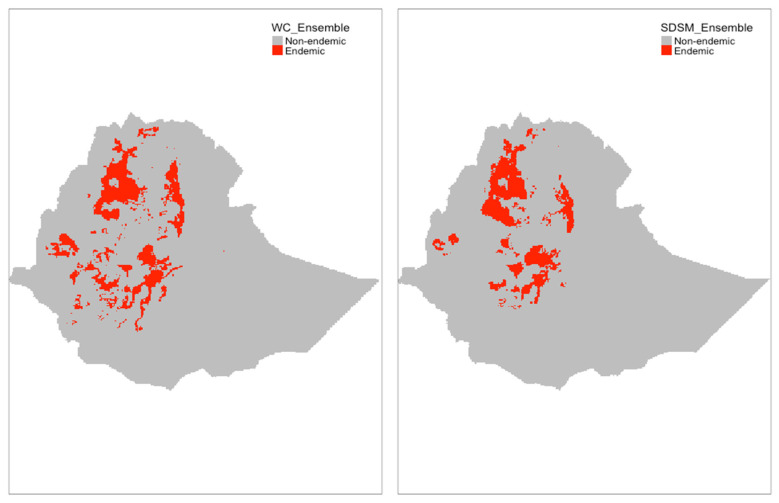
Distribution of *S. mansoni* endemic areas in Ethiopia projected by WC-based (**left**) and SDSM-based (**right**) ensemble models. An area is ‘endemic’ if the model-predicted probability of presence is ≥0.65 and considered ‘nonendemic’ otherwise. The threshold was chosen to maximize balance accuracy based on the ground-truthing data.

**Table 1 microorganisms-09-02144-t001:** Environmental and ecological variables, and data sources.

Data	Unit	Resolution	Time Period	Source
Elevation	Meters Above Sea Level	250 m		CGIAR Consortium for Spatial Information (CGIAR-CSI) [20]
Wealth Index Score		250 m	2016	Demographic and Health Surveys (DHS) [23]
Silt	g/100 g	250 m	2008–2012	International Soil Reference and Information Centre (ISRIC) [28]
Sand	g/100 g	250 m	2008–2012	International Soil Reference and Information Centre (ISRIC) [28]
Clay	g/100 g	250 m	2008–2012	International Soil Reference and Information Centre (ISRIC) [28]
NDVI *		1 km	1999–2017	Copernicus Global Land Service [21]
Mean Temperature	Degrees Celsius	1 km	1970–2000	WorldClim [29]
Annual Precipitation	mm	1 km	1970–2000	WorldClim [29]
Max Temperature	Degrees Celsius	250 m	1961–2005	Gebrechorkos SH et al. [30]
Min Temperature	Degrees Celsius	250 m	1961–2005	Gebrechorkos SH et al. [30]
Mean Precipitation	mm	250 m	1961–2005	Gebrechorkos SH et al. [30]

* NDVI = Normalized Difference Vegetation Index.

**Table 2 microorganisms-09-02144-t002:** Testing AUC values of ENM models based on WC and SDSM datasets.

Model	WC		SDSM	
	AUC	Rank	AUC	Rank
GLM	0.883	5	0.851	6
GBM	0.902	3	0.879	3
GAM	0.879	6	0.848	7
SRE	0.723	9	0.735	9
CTA	0.788	8	0.798	8
RF	0.929	2	0.875	4
MARS	0.871	7	0.880	2
MAXENT	0.889	4	0.872	5
Ensemble	0.956	1	0.960	1

**Table 3 microorganisms-09-02144-t003:** Field ground-truthing validation using the accuracy (ACC), sensitivity (SE) specificity (SP), positive predictive value (PPV), negative predictive value (NPV), and balanced accuracy (BA).

Model	WC						SDSM					
	ACC (%)	SE (%)	SP (%)	PPV (%)	NPV (%)	BA (%)	ACC (%)	SE (%)	SP (%)	PPV (%)	NPV (%)	BA (%)
GLM	58.33	75.00	25.00	69.57	100.00	50.00	50.00	43.75	62.50	69.57	100.00	53.13
GAM	70.83	93.75	25.00	71.43	66.67	59.38	45.83	56.25	25.00	71.43	66.67	40.63
MARS	66.67	75.00	50.00	80.00	100.00	62.00	45.83	43.75	50.00	80.00	100.00	46.88
CTA	75.00	100.00	25.00	72.73	100.00	62.50	79.17	100.00	37.50	72.73	100.00	68.75
RF	41.67	43.75	37.50	84.21	100.00	40.63	83.33	81.25	87.50	84.21	100.00	84.38
GBM	83.33	93.75	62.50	78.95	80.00	78.13	58.33	50.00	75.00	78.95	80.00	62.50
SRE	54.17	75.00	12.50	66.67	33.33	43.75	58.33	75.00	25.00	66.67	33.33	50.00
MAXENT	37.50	25.00	62.50	81.82	46.15	43.75	29.17	12.50	62.50	81.82	46.15	37.50
Ensemble	83.33	93.75	62.50	84.21	100.00	78.13	41.67	31.25	62.50	84.21	100.00	46.88

## Data Availability

The dataset will be available in the following repositories (Open Science Framework (https://osf.io)) or upon request to the authors.

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
