# Peer review of "Environmental Drivers and Potential Distribution of Schistosoma mansoni Endemic Areas in Ethiopia"

_microorganisms, 2021, doi:10.3390/microorganisms9102144_

Round 1

Reviewer 1 Report

This manuscript aims to propose a modeling approach in order to map the schistosomiasis endemic areas and inform public health intervention and surveillance, which is really interesting.

My main concern is about the parameters used in the models. The authors only used the metrics accuracy, sensitivity and specificity; they should add in their analyses the following three metrics: Proportion of true positive (PPV), Proportion of true negative (NPV) and balanced accuracy which is the arithmetic mean of the two metrics (sensitivity and specificity), and a powerful and useful metric for field data. They should redo their maps based on the ensemble balanced accuracy for both WC and SDSM and also explain the numbers in Figures 3 and 4.

Author Response

Reviewer1: This manuscript aims to propose a modeling approach in order to map the schistosomiasis endemic areas and inform public health intervention and surveillance, which is really interesting.

R1: My main concern is about the parameters used in the models. The authors only used the metrics accuracy, sensitivity and specificity; they should add in their analyses the following three metrics: Proportion of true positive (PPV), Proportion of true negative (NPV) and balanced accuracy which is the arithmetic mean of the two metrics (sensitivity and specificity), and a powerful and useful metric for field data.

Response: Thanks for the overall positive comments. Following the reviewer’s suggestion, we calculate the three more metrics (PPV, NPV and balance accuracy) into the field ground truthing validation, as shown in the table below. As you can see, the newly added metrics do not add extra values to the study; on contrary, the balance accuracy (BC) even performs less well than the accuracy (ACC). We reviewed this situation thoroughly and believe that this is due to the fact that our field dataset is overall quite balanced, in contrast to the situation that BC is commonly used in imbalance dataset, like many other applications do.

Table The field ground truthing validation

Model

WC

SDSM

ACC (%)

Se

(%)

Sp

(%)

PPV (%)

NPV (%)

BA

(%)

ACC

(%)

Se

(%)

Sp

(%)

PPV

(%)

NPV

(%)

BA

(%)

GLM

70.83

100.00

12.50

69.57

100.00

56.25

41.67

50.00

25.00

57.14

20.00

37.50

GAM

70.83

93.75

25.00

71.43

66.67

59.38

62.50

81.25

25.00

68.42

40.00

53.13

MARS

83.33

100.00

50.00

80.00

100.00

75.00

41.67

50.00

25.00

57.14

20.00

37.50

CTA

75.00

100.00

25.00

72.73

100.00

62.50

79.17

100.00

37.5

76.19

100.00

68.75

RF

87.50

100.00

62.50

84.21

100.00

81.25

83.33

87.50

75.00

87.50

75.00

81.25

GBM

79.17

93.75

50.00

78.95

80.00

71.88

79.17

87.50

62.50

82.35

71.43

75.00

SRE

62.50

87.50

12.50

66.67

33.33

50.00

25.00

31.25

12.50

41.67

8.33

21.88

MAXENT

62.50

56.25

75.00

81.82

46.15

62.50

37.50

25.00

62.50

57.14

29.41

43.75

Ensemble

87.50

100.00

62.50

84.21

100.00

81.25

41.67

43.75

37.50

58.33

25.00

40.63

R1: They should redo their maps based on the ensemble balanced accuracy for both WC and SDSM and also explain the numbers in Figures 3 and 4.

Response: We added more descriptions regarding the number in the figure 3 and 4 (Page 10 and 11)

Figure 3 WC-based results showing potential distribution of S. mansoni endemic areas in Ethiopia from different models. Blue dots represent the occurrence locations. Legend shows a probability of occurrence given the environmental suitability ranging 0 to 100 percent.

Figure 4 SDSM-based results showing potential distribution of S. mansoni endemic areas in Ethiopia from different models. Blue dots represent the occurrence locations. Legend shows a probability of occurrence given the environmental suitability ranging 0 to 100 percent.

Reviewer2: This paper intend to relate environmental factors to the presence of Shistosoma mansoni in different regions of Ethiopia. They choose ecological niche modelling to analyze their data.

R2: I did not see clearly if they use data modelling (the model fit well the data) or algorithmic modelling (the model predicts well on other data) as explained in DL Warren and SN Seifert Ecological niche modelling in Maxent. Ecological Applications, 2011.

Response: Thanks for rising the point on the lack of clarity in the method description. In the modeling exercises performed in the study, we used out-of-sample validation approach or algorithmic modeling approach, as the reviewer pointed out. Across all modeling techniques used, we employed the 80:20 split scheme for model training and validation (e.g., 80% of data are randomly chosen to construct the model, while the rest 20% to evaluate the predictive values of the model), respectively. We have further clarified this in the revision text (Page 5 line 178 - 181)

“For the former, all occurrence (95 sites) and pseudoabsence (1,000 sites) data were randomly split into two parts with an 80:20 ratio, with 80% (training set) for training the model and the remaining 20% (testing set) for validating the predictive performance of the model.

R2: They build the model on 80% of data and predictd the rest of 20% of these data so it appears it rather algorithmic modelling. Could they clarify?

Response: Again, thanks for the good point and sorry for the lack of clarity. Yes, we employed the algorithmic modeling and please see the response above.

R2: They had 92 sites with presence and generated 1000 pseudoabsence data, that were included in the model in order to obtain a better fit/prediction. It seems then that most of the analyzed data are randomly generated and then you may be skeptic on the value of the model. Why 1000 generated pseudoabsence and not 100 or 50?

R2: The generated sites of pseudoabsence where were located at 20 km from positive sites; why?

Response: It is a common practice that many niche modeling techniques use pseudoabsence data to augment the actual ‘absence’ data points. There are theoretical and practical considerations behind the practice. The number of pseudoabsence data points and radius range chosen in practice are largely based on empirical evidence from some previous studies (see reference below), which suggested that the model accuracy is usually maximized when proportion of presence and pseudoabsence reach approximately 10 percent prevalence. We have further clarified this in the revision by adding the following text (Page 2-3, line 89-100). 

“To follow widely accepted practices in the ecological field, we augmented the presence data with pseudoabsence data points for the niche modeling [17, 18]. Specifically, 1,000 pseudoabsence data points in 20 kilometers away from the ‘occurrence’ locations were randomly generated across the country. The number of the pseudoabsence data points is based on empirical evidence from a previous study [19] which suggests that the model accuracy is usually satisfactory when the ratio of presence vs. pseudoabsence reaches approximately 1:10. Choosing an area of 20 km radius to generate pseudoabsence points is based on the resolution of environmental data (1 km2), allowing adequate environmental differentiation between presence and pseudoabsence locations, an criterion that has also been empirically confirmed by the field epidemiologist and parasitologist in Ethiopia (BE). The procedure of selecting pseudoabsence points was implemented in ArcGIS version 10.3.1.

R2: I did not clearly understood the fit (or rather the quality of predictions) on the 20% of the remaining data. Could they explain more? Was it on the 1000+92 sites?

Response: Thanks again and sorry for unclear explanation. The revision was made and please see the response above.

R2: why not use standard GLM methods with prevalences? Probably they would need real absence data for this, which could be found probably.

Response: Thanks for the valuable point. In this study, our focus was on suitability of areas for the S. mansoni transmission or not, therefore we were interested in a binary outcome of the study area – yes (presence) or no (absence). Considering the level of transmission (e.g., measured through prevalence of infection or infection intensity) in the niche modeling framework is our next step exploration.

Reviewer 2 Report

This paper intend to relate environmental factors to the presence of Shistosoma mansoni in different regions of Ethiopia.  They choose ecological niche modelling to analyze their data.

I did not see clearly  if they use data modelling (the model fit well the data) or algorithmic modelling (the model predicts well on other data) as explained in DL Warren and SN Seifert Ecological niche  modelling in Maxent.. Ecological Applications, 2011. They build the model on 80 ù of data and predictd the rest of 20% of these data so it appears it rather algorithmic modelling. Could they clarify?

They had 92 sites with presence and generated 1000 pseudoabsence data, that were included in the model in order to obtain a better fit/prediction. It seems then that most of the analyzed data are randomly generated and then you may be skeptic on the value of the model. Why 1000 generated pseudoabsence and not 100 or 50? The generated sites of pseudoabsence where were located at 20 km from positive sites; why?

I did not clearly understood the  fit (or rather the quality of predictions) on the 20% of  the remaining data. Could they explain more? Was it on the 1000+92 sites?

My last question is : why not use standard GLM methods with prevalences ? Probably they would need real absence data for this, which could be found probably.

Otherwise, the subject is interesting and may warrant publication if the questions are answered.

Author Response

Reviewer2: This paper intend to relate environmental factors to the presence of Shistosoma mansoni in different regions of Ethiopia. They choose ecological niche modelling to analyze their data.

R2: I did not see clearly if they use data modelling (the model fit well the data) or algorithmic modelling (the model predicts well on other data) as explained in DL Warren and SN Seifert Ecological niche modelling in Maxent. Ecological Applications, 2011.

Response: Thanks for rising the point on the lack of clarity in the method description. In the modeling exercises performed in the study, we used out-of-sample validation approach or algorithmic modeling approach, as the reviewer pointed out. Across all modeling techniques used, we employed the 80:20 split scheme for model training and validation (e.g., 80% of data are randomly chosen to construct the model, while the rest 20% to evaluate the predictive values of the model), respectively. We have further clarified this in the revision text (Page 5 line 178 - 181)

“For the former, all occurrence (95 sites) and pseudoabsence (1,000 sites) data were randomly split into two parts with an 80:20 ratio, with 80% (training set) for training the model and the remaining 20% (testing set) for validating the predictive performance of the model.

R2: They build the model on 80% of data and predictd the rest of 20% of these data so it appears it rather algorithmic modelling. Could they clarify?

Response: Again, thanks for the good point and sorry for the lack of clarity. Yes, we employed the algorithmic modeling and please see the response above.

R2: They had 92 sites with presence and generated 1000 pseudoabsence data, that were included in the model in order to obtain a better fit/prediction. It seems then that most of the analyzed data are randomly generated and then you may be skeptic on the value of the model. Why 1000 generated pseudoabsence and not 100 or 50?

R2: The generated sites of pseudoabsence where were located at 20 km from positive sites; why?

Response: It is a common practice that many niche modeling techniques use pseudoabsence data to augment the actual ‘absence’ data points. There are theoretical and practical considerations behind the practice. The number of pseudoabsence data points and radius range chosen in practice are largely based on empirical evidence from some previous studies (see reference below), which suggested that the model accuracy is usually maximized when proportion of presence and pseudoabsence reach approximately 10 percent prevalence. We have further clarified this in the revision by adding the following text (Page 2-3, line 89-100). 

“To follow widely accepted practices in the ecological field, we augmented the presence data with pseudoabsence data points for the niche modeling [17, 18]. Specifically, 1,000 pseudoabsence data points in 20 kilometers away from the ‘occurrence’ locations were randomly generated across the country. The number of the pseudoabsence data points is based on empirical evidence from a previous study [19] which suggests that the model accuracy is usually satisfactory when the ratio of presence vs. pseudoabsence reaches approximately 1:10. Choosing an area of 20 km radius to generate pseudoabsence points is based on the resolution of environmental data (1 km2), allowing adequate environmental differentiation between presence and pseudoabsence locations, an criterion that has also been empirically confirmed by the field epidemiologist and parasitologist in Ethiopia (BE). The procedure of selecting pseudoabsence points was implemented in ArcGIS version 10.3.1.

R2: I did not clearly understood the fit (or rather the quality of predictions) on the 20% of the remaining data. Could they explain more? Was it on the 1000+92 sites?

Response: Thanks again and sorry for unclear explanation. The revision was made and please see the response above.

R2: why not use standard GLM methods with prevalences? Probably they would need real absence data for this, which could be found probably.

Response: Thanks for the valuable point. In this study, our focus was on suitability of areas for the S. mansoni transmission or not, therefore we were interested in a binary outcome of the study area – yes (presence) or no (absence). Considering the level of transmission (e.g., measured through prevalence of infection or infection intensity) in the niche modeling framework is our next step exploration.

Round 2

Reviewer 1 Report

The authors responded: As you can see, the newly added metrics do not add extra values to the study; on contrary, the balance accuracy (BC) even performs less well than the accuracy (ACC). We reviewed this situation thoroughly and believe that this is due to the fact that our field dataset is overall quite balanced, in contrast to the situation that BC is commonly used in imbalance dataset, like many other applications do.

The authors did not want to include the new metrics I asked arguing that they believe that their field dataset is overall quite balanced. However, they did not prove that their field dataset was balanced. I suggest that the authors provide in their manuscript a statistical test to prove the balanced dataset; if not, they should use the new metrics I already proposed, change their figures and have a true discussion on their results.

Reviewer 2 Report

The  paper has been clearly improved particularly on the methodology. It is clearer than the previous version. It is ready for publication.

Author Response

Thank you for all valuable comments